# Functional Identification and Genetic Transformation of the Ammonium Transporter *PtrAMT1;6* in *Populus*

**DOI:** 10.3390/ijms24108511

**Published:** 2023-05-10

**Authors:** Chengjun Yang, Chunxi Huang, Luzheng Gou, Han Yang, Guanjun Liu

**Affiliations:** 1Northeast Asia Biodiversity Research Center, Northeast Forestry University, Harbin 150040, China; nxyycj@nefu.edu.cn (C.Y.); huangchunxi@nefu.edu.cn (C.H.); 2State Key Laboratory of Tree Genetics and Breeding, Northeast Forestry University, Harbin 150040, China

**Keywords:** nitrogen, ammonium transporter, *P. trichocarpa*, yeast functional complementation, genetic transformation, *PtrAMT1;6*

## Abstract

The ammonium transporter (AMT) family gene is an important transporter involved in ammonium uptake and transfer in plants and is mainly engaged in the uptake and transport of ammonium from the environment by roots and the reabsorption of ammonium in the aboveground parts. In this study, the expression pattern, functional identification, and genetic transformation of the *PtrAMT1;6* gene, a member of the ammonium transporter protein family in *P. trichocarpa*, were investigated as follows: (1) Fluorescence quantitative PCR demonstrated that the *PtrAMT1;6* gene was preferentially expressed in the leaves, with both dark-induced and light-inhibited expression patterns. (2) A functional restoration assay using the yeast ammonium transporter protein mutant strain indicated that the *PtrAMT1;6* gene restored the ability of the mutant to transport ammonium with high affinity. (3) *Arabidopsis* was transformed with pCAMBIA-*PtrAMT1;6P,* and the transformed lines were stained with GUS, which showed that the rootstock junction, cotyledon petioles, and the leaf veins and pulp near the petioles of the transformed plants could be stained blue, indicating that the promoter of the *PtrAMT1;6* gene had promoter activity. (4) The overexpression of the *PtrAMT1;6* gene caused an imbalance in carbon and nitrogen metabolism and reduced nitrogen assimilation ability in ‘84K’ poplar and ultimately reduced biomass. The above results suggest that *PtrAMT1;6* may be involved in ammonia recycling during nitrogen metabolism in aboveground parts, and overexpression of *PtrAMT1;6* may affect the process of carbon and nitrogen metabolism, as well as nitrogen assimilation in plants, resulting in stunted growth of overexpression plants.

## 1. Introduction

Nitrogen is one of the most abundant nutrients in living organisms and is essential for plant growth and development; nitrogen nutrients constitute 70% of the nutrients obtained by plants [1]. However, almost every plant faces a nitrogen nutrient deficiency at some point during its growth and development [2]; in fact, nitrogen nutrition is the most important limiting factor for plant growth and development and can limit plant growth by more than 50%, resulting in reduced yields of crops and forest trees [3]. The main nitrogen nutrients contained in soil include inorganic nitrogen, such as ammonium and nitrate; and organic nitrogen, such as amino acids, peptides, and proteins, of which inorganic nitrogen serves as the most important form of nitrogen nutrient uptake by plants [4,5]. Plants can efficiently use ammonium transporters (*AMTs*) and nitrogen transporters (*NRTs*) to absorb inorganic nitrogen; the efficiency of absorption is much higher than that of organic nitrogen, and the energy required to absorb ammonium nitrogen(NH_3_/NH_4_^+^) is much less than that required to absorb nitrate nitrogen (NO_3_^2212^ /NO_2_^2212^), so plants use NH_3_/NH_4_^+^ in the most economical way, and for many plants, NH_3_/NH_4_^+^ is their preferred nitrogen [6]. However, too much ammonium in the cell can inhibit plant growth or even produce pressures, so the concentration of ammonium in the intracellular environment is strictly regulated by the plant [7,8]. Poplar trees only need a small amount of N to grow and develop normally and become good wood. The forest trees do not compete with crops for farmland, and research on transgenic woody plants is on the agenda.

To date, there have been multiple reports on the cloning and expression analysis of *AMT* genes in *Arabidopsis*, rice, and tomato. Six of these members are *AMT1;1*-*AMT1;5* and *AMT2* in *Arabidopsis* [9]. The expression of *AMT1;1* in *Arabidopsis* is repressed by glutamine [1,10], and *AMT1;3* has a specific phosphorylation site at the C-terminus, a putative receptor for ammonium signaling with the function of regulating lateral root genesis in response to the ammonium environment [11,12].

The high-affinity *AMT* proteins in rice are *OsAMT1;1*, *OsAMT1;2*, and *OsAMT1;3*: *OsAMT1;1* is constitutively expressed in the aboveground parts and ammonium-induced in the belowground parts; *OsAMT1;2* is root-specific and ammonium-induced; and *OsAMT1;3* is root specific, and its expression is influenced by nitrogen in the root (ammonium or nitrate nitrogen) inhibition [13].

There are three members of the high-affinity *AMT* protein gene family in tomato, *LeAMT1;1*, *LeAMT1;2*, and *LeAMT1;3* [14]. All three members are able to restore ammonium uptake in yeast mutants [5]. *LeAMT1;1* is constitutively expressed, while *LeAMT1;2* and *LeAMT1;3* respond to light periodicity but have opposite expression patterns [15], with *LeAMT1;2* being highly expressed in the light, where it absorbs ammonium emitted in the plastid ectodomain due to photorespiration, and *LeAMT1;3* being highly expressed in the dark, where it mainly absorbs ammonium due to transamination or deamination [16,17].

To date, researchers have identified a total of seven high-affinity *AMT* proteins, *PtrAMT1;1*-*PtrAMT1;7*, in the poplar genome [18]. Couturier et al. not only performed photoperiodic analysis and partial tissue-specific expression analysis of the AMT family on *P. trichocarpa* but also cloned ten genes, including *AMT1;1*-*AMT1;6*, from *Populus tremula* × *alba* (clone INRA 717 1B4), and the heterologous expression of *AMT1;2* and *AMT1;6* in yeast on their functions were studied in detail [5]. On this basis, we not only performed whole-tissue and photoperiodic expression pattern analysis in order to further explore the function of *PtrAMT1;6* gene, but also cloned the *PtrAMT1;6* gene from *P. trichocarpa* and determined the function of this gene by yeast mutant function recovery assay. In addition to this, we fused the PtrAMT1;6 gene promoter with the GUS gene to explore the function of the promoter. We also overexpressed the PtrAMT1;6 gene in *Populus alba* × *P. glandulosa* cv. ‘84K’ by the Agrobacterium-mediated genetic transformation method and suppressed the endogenous homologous gene in ‘84K’ poplar under three NH_4_Cl concentrations (0.1 mM, 2 mM and 10 mM) to analyze its growth phenotype, carbon and nitrogen metabolism status and gene expression, and other characteristics to lay the foundation for the further study of gene function.

## 2. Results

### 2.1. Expression Pattern of the P. trichocarpa Ammonium Transporter

The *PtrAMT1;6* gene is mainly expressed in leaves and may be involved in the uptake of ammonium produced by metabolism in the leaves; the transcript level of the *PtrAMT1;6* gene is higher in mature leaves, petioles, and stems than in young parts, and catabolism in mature leaves is higher than that of young leaves, so we speculate that the function of the *PtrAMT1;6* gene may be involved in the metabolism of amino acids to yield ammonium reabsorption; the gene maintains a certain level of expression in stems and petioles, suggesting that it may be involved in the translocation of ammonium produced by the metabolism of phenylpropanoids (Figure 1A).

Since the *PtrAMT1;6* gene is mainly expressed in leaves, three fully expanded mature leaves were selected as targets for examining its photoperiod regulation. To study the photoperiodical expression pattern of the PtrAMT1;6 gene, in addition to selecting the time points under light and dark conditions, we also selected the light/dark and dark/1 h before the introduction of the light and the point of intersection (Figure 1B). After initial exposure to light (6:00) for 1 h (7:00), the transcription level of *PtrAMT1;6* was strongly inhibited, and the expression level was reduced by 46%, while after the transition from the light (22:00) to the dark (23:00), the transcription level of *PtrAMT1;6* was strongly induced, increasing by 53%. It is speculated that the gene may engage in amino acid metabolism in the dark and ammonium transport in the GDH pathway. At 10:00, after the plant had been in the light for 4 h, the expression level of *PtrAMT1;6* was 42% higher compared with 7:00. This phenomenon may mean that *PtrAMT1;6* is also involved in ammonium recovery during photorespiration. The overall expression of the gene in the dark was 50% higher than the overall expression in the light, indicating that the gene was mainly expressed in the dark.

### 2.2. Multiple Alignment and Transmembrane Domain Prediction of the N-Terminal Sequence of the AMT1 Protein Family in Populus Trichocarpa and Solanum lycopersicum L.

The protein N-terminal sequences of the *AMT1* family members of *P. trichocarpa* and *Solanum lycopersicum* L. were subjected to multiple sequence alignment [19] (Figure 2A). *PtrAMT1;6* and *LeAMT1;3* have shorter N-termini and high homology, and both lack two cysteines that are important for stabilizing homologous trimers (black dots in Figure 2A). Information on transmembrane helices was predicted with the online software TMHMM (Figure 2B). The PtrAMT1;6 protein has nine putative transmembrane helices. Of the other *AMT1* homologues in *P. trichocarpa* [18], PtrAMT1;4 and PtrAMT1;7 have 10 and 7 putative transmembrane helices, respectively, and the others all have 9 transmembrane helices; PtrAMT1;6 has a short N-terminus outside the cell membrane. It can be speculated by sequence analysis that PtrAMT1;6 is a membrane protein with 11 putative transmembrane helices and has high homology with the LeAMT1;3 protein N-terminal sequence, and both are relatively short. These results elucidate the structural and functional homology of the two genes. Meanwhile, due to its missing N-terminal sequence, whether PtrAMT1;6 has an ammonium transport function is the premise of further study.

### 2.3. Yeast Complementary Function

To characterize the function of the *PtrAMT1;6* gene, the pYES2-*PtrAMT1;6* vector was constructed and used in a functional recovery assay with the yeast ammonium transporter protein triple-deletion mutant *31019b* (Figure 3). Yeast strain *31019b* (*mep1, mep2, mep3*) is a high-affinity ammonium transport system deletion mutant, so it lacks the high-affinity ammonium absorption ability and cannot grow at an ammonium concentration below 5 mM. *31019b*-pYES2-*PtrAMT1;6* can be grown at 0.4 mM and 1 mM ammonium concentrations, indicating that the *P. trichocarpa* transporter member *PtrAMT1;6* has high affinity for ammonium absorption. At 20 mM and 100 mM ammonium concentrations, strains that contain no recombinant plasmid can grow because of the functions of the low-affinity ammonium transport system, but due to its inefficiency, ammonium toxicity is not possible. Moreover, *31019b*-pYES2-*PtrAMT1;6* could not survive at these two ammonium concentrations because the high-affinity ammonium absorption function of the mutant was restored by *PtrAMT1;6*, resulting in toxicity due to the absorption of large amounts of ammonium ions. The strain containing no recombinant plasmid was able to survive at any of the above methylammonium (MEA) concentrations because the MEP-type ammonium transporter present in the yeast does not have the high affinity of MEA, so it does not produce a toxic reaction, and *31019b*-pYES2-*PtrAMT1;6* cannot grow, indicating that *PtrAMT1;6* has the ability to bind to transport MEA and is thus an AMT-type plant ammonium transporter.

### 2.4. Arabidopsis GUS Staining

#### 2.4.1. Promoter Component Analysis

Primers were designed based on the genome sequence of *P. trichocarpa*, and the PtrAMT1;6 promoter was cloned. The sequencing results are consistent with the *P. trichocarpa* database. An analysis of the promoter revealed that the promoter region contained a photoresponsive element, an ethylene response element, a rhythm control element, an anoxic sensing element, and others. Important elements in the promoter include 9 light-response-related elements, 1 ethylene response element, 2 circadian control elements, and 2 anaerobic induction elements (Table 1).

#### 2.4.2. Expression of PtrAMT1;6P::GUS in Arabidopsis

The plant expression vector pCAMBIA 1301-*PtrAMT1;6*P was constructed, and *Arabidopsis* was genetically transformed and GUS stained. The results are shown in Figure 4. At the seedling stage, GUS was mainly expressed on the veins of the rhizome, the cotyledon petioles, and the veins and mesophyll cells at the close junction with the petiole but not in the roots or in the newly emerged leaves. Although the morphological structures of *Arabidopsis* and *P. trichocarpa* are very different, the expression activity of the *PtrAMT1;6* promoter in *Arabidopsis* cannot fully explain its expression activity in *P. trichocarpa*, but this result indicates that the *PtrAMT1;6* promoter has promoter activity; GUS is not expressed in the root, suggesting that PtrAMT1;6 may not play a major role in the root.

### 2.5. Overexpression of PtrAMT1;6 in Populus alba × P. glandulosa cv. ‘84K’

#### 2.5.1. *PtrAMT1;6* Gene Expression Detection

To further investigate the function of *PtrAMT1;6* in ammonium transport and uptake, we constructed ‘84k’ poplar overexpression lines, as well as suppressor lines, and examined gene expression. The results (Figure 5) showed that the relative expression of the *PtrAMT1;6* gene in the 12 overexpression lines obtained was higher than that of wild-type(wt) plants, with o2 expression being the highest, 48 times higher than that of the wild type; o1, o3, o4, o17, and o19 being 15, 39, 32, 24, and 19 times higher than that of the wild type, respectively; and o15 expression being lower, only twice as high as that of the wild type. The expression of the suppressor strains was lower than that of the wt plants, with i5 being the lowest, less than 40% of the wild type.

#### 2.5.2. Growth Phenotype of Outdoor Potted ‘84K’ poplar

Plant height and ground diameter were measured every 20 days, starting from transplanting to the greenhouse, and the increase was measured four times. The height growth of the wt and suppressor strains was significantly higher than that of the overexpression strain during the 2-month test period. The growth rates of o1, o16, and o17 in the overexpression strain were faster than those of i3 and i5 in the suppression strain. The increase in ground diameter from May 10 to July 10 (two months) showed that the overexpression strains o3, o4, o6, o9, and o22 grew slower than the wild type, while the suppression strains i3 and i5 grew faster. Overall, the growth of overexpression plants under outdoor growth conditions was significantly lower than that of the suppressor strains and wt plants (Figure 6).

From these, we selected the differentially expressed overexpression lines o2 and o4 and the suppressor lines i5 and i10 for the following nitrogen treatment experiments. Due to individual differences in seedlings, the initial plant height and ground diameter were measured before starting the nitrogen treatment to ensure the accuracy of the test, and then every other week, this was performed three times to calculate the increases in plant height, root length, and dry weight (DW). The results indicated (Figure 7) that the increase in plant height first increased and then decreased with the increasing nitrogen concentration in the wt and suppressor strains and decreased with the increasing nitrogen in the overexpress or strains, with the wt plants being taller than the suppressor strain at 0.1 mM and 2 mM and the opposite at 10 mM. Increases in ground diameter were generally consistent with the changes in plant height. The root length of the overexpression/suppression strains was lower than that of the wt plants under the three nitrogen conditions, with significant differences. For dry weight, there was no difference between the wt and suppressor strains at 0.1 mM, and the wild type had a greater dry weight than the suppressor strains at 2 mM and 10 mM, with significant differences. The overall results showed that the increases in plant height and ground diameter, as well as the root length and dry weight of the two overexpression lines, were significantly lower than those of the wt and suppressor lines at the three nitrogen concentrations.

#### 2.5.3. Photosynthetic Parameters

Table 2 indicates that the change trend of the SPAD value, photosynthetic rate, and intercellular carbon dioxide concentration in the soil growth conditions of the greenhouse was consistent. The values for all but one or two of the 13 overexpression strains were lower than those of the suppression strain and the wt plant but showed the opposite trend in terms of stomatal conductance and transpiration rate. The overexpression lines showed higher performance, and the suppression lines and wt plants showed lower performance.

Table 3 shows that as the nitrogen application level increased, the SPAD value and photosynthetic rate of the plants increased in both transgenic and wt plants. In the three nitrogen concentrations tested, except for the intercellular carbon dioxide concentration and transpiration rate at 10 mM, which was slightly lower than that of the i10 strain, the wt plants had higher SPAD values and photosynthetic rates than the transgenic lines.

#### 2.5.4. Total Carbon and Nitrogen Content and Carbon-to-Nitrogen Ratio

To investigate the accumulation of nitrogen in ‘84K’ poplar, we treated the wild type and the four transformed strains with three nitrogen concentrations for 15 d and then measured the total carbon and total nitrogen content in the roots, stems, and leaves to calculate the carbon-to-nitrogen ratio. The results indicated (Figure 8) that the total nitrogen content increased with the increasing nitrogen concentration in roots, stems, and leaves, and the total nitrogen content was higher in leaves than in roots and stems. The total nitrogen content in the leaves of the overexpression strains was significantly higher than that of both the wt and suppressor strains. In the stems, the wt content was highest at 0.1 mM and 10 mM, and the content in overexpression strains was higher than that in the suppressor strains; at 2 mM, the content in o2 was highest, and the content in the suppressor strains was lower than that of the wt and overexpression strains. The total nitrogen content in the roots of the two overexpression lines was significantly lower than that of the wt and suppressor lines. The total plant carbon content was less different among the strains under different nitrogen conditions. The carbon-to-nitrogen ratio of the plants decreased with increasing nitrogen concentration, which was mainly due to the corresponding increase in total nitrogen content and smaller change in total carbon content after increasing the nitrogen concentration. The overexpression strains showed a lower carbon-to-nitrogen ratio in the leaves at all three nitrogen concentrations, and a higher carbon-to-nitrogen ratio was observed in the wt; the overexpression strains showed a lower carbon-to-nitrogen ratio in the stems than the wt and suppressor strains at 0.1 mM and 2 mM, with the lowest ratio in the wt and the highest in the suppressor strains at 10 mM; the changes in the roots were consistent with those in the stems.

#### 2.5.5. Soluble Protein and Soluble Sugar Content

Soluble protein is the main product of nitrogen anabolism, and its content reflects the assimilation capacity of plants for nitrogen; soluble sugar is the primary product of photosynthetic carbon assimilation, and its content marks the supply capacity of assimilates in plants and reflects the conversion and utilization capacity of plants for assimilates. To examine whether overexpression of *PtrAMT1;6* in transgenic plants caused changes in soluble protein and soluble sugar contents, we examined the overexpression and suppression lines and the wt ‘84K’ poplar grown for 15 d at different nitrogen concentrations. The overexpression lines o2 and o4 had higher soluble protein content in the leaves (Figure 9A) than the wt, i5, and i10 under 0.1 mM conditions, with i10 having the lowest content; the situation was reversed under 2 mM conditions, with no difference between wt, o2, and i10, and o4 having the lowest content. The differences among the lines under 10 mM conditions were small; in the stems, the overexpression lines under the three nitrogen concentration conditions showed lower soluble protein content than the suppression strains and the wt. The soluble protein content of wt was higher than that of both the overexpression and suppression strains; in the roots, o4 had the lowest content at 0.1 mM and differed significantly from the other strains; wt was the highest at 2 mM and 10 mM, followed by the overexpression strains, and the suppression strains showed the lowest content. Overall, the soluble protein content of all organs of the plant increased with increasing nitrogen concentration and was highest in the leaves and lowest in the roots.

The content of soluble sugars (Figure 9B) increased with increasing nitrogen concentration in the leaves and stems and decreased with increasing nitrogen concentration in the roots and were higher in the leaves than in the stems and roots and lowest in the roots; the content of the leaves of the wt was significantly higher than that in other strains under 2 mM and 10 mM conditions, with the highest content in o2 and the lowest content in i10 under 0.1 mM conditions. In the stems, the differences among strains were small; in the roots, wt showed higher levels at all three nitrogen concentrations, with lower levels in the overexpression strains.

#### 2.5.6. Glutamine Synthetase Content

Glutamine synthetase (GS) is mainly found in plants and is a key enzyme for ammonium assimilation in organisms. It is also the main form of storage and transportation of ammonium. To determine whether overexpressed *PtrAMT1;6* in transgenic plants caused changes in glutamine synthetase content, we tested this indicator in the overexpression and suppression lines and wt ‘84K’ poplar grown for 15 days at different nitrogen concentrations (Figure 10). In general, the glutamine synthetase content increased with the increasing nitrogen concentration, but the content in the leaves of the overexpression strains o2 and o4 at 0.1 mM was significantly higher than that of the wt and suppression strains; the content was markedly lower at 2 mM and higher at 10 mM. It is worth noting that the content in i5 was at a high level under all three nitrogen concentrations. The contents in the root in each strain were essentially the same under the three nitrogen concentrations, with the contents in wt and i10 being higher and the content of i5 being the lowest.

#### 2.5.7. Free Amino Acid Content

The amino acid content in plants is important for studying the changes in nitrogen metabolism in plants, plant nitrogen absorption, transportation, assimilation, and nutritional status under different conditions. To determine whether the overexpression of PtrAMT1;6 in transgenic plants caused a change in free amino acid content, we tested this indicator in the overexpression and suppression strains and wt ‘84K’ poplar grown for 15 d at different nitrogen concentrations (Figure 11). The results showed that the leaves of the i5 strain had higher amino acid content under the three nitrogen concentrations, and the content in the overexpression strains was relatively low at the 2 mM concentration; the content of the roots of the two overexpression lines was lower than that of the suppression lines and WT.

#### 2.5.8. Relative Expression Levels of the Ammonium Transporter and Nitrate Transporter Genes

The expression of the *PtrAMT1;6* gene in the leaves and the expression of the *NRT1;1* gene in the roots were examined because the transporter protein was expressed in the middle of the root (Figure 12). The *PtrAMT1;6* gene in the leaves was expressed at ultrahigh levels in the two overexpression lines at the three nitrogen concentrations, more than four-fold higher than in the wt and suppressor lines, and the expression of the suppressor lines i5 and i10 were lower than that of wt. The expression of each line increased with the increasing nitrogen concentration. The expression of the *NRT1.1* gene in the roots was highest in the o2 strain at 0.1 mmol/L nitrogen concentration, but the difference between the lower expression of o4 and that of the suppressor strains i5 and i10 was not significant; the expression in o4 and i10 was higher at the 2 mmol/L nitrogen concentration, and the difference between wt, o2, and i5, which had lower expression, was not significant; the expression of i5 was higher at 10 mmol/L nitrogen, and the difference between wt, o2, o4, and i10 was not significant.

#### 2.5.9. Relative Expression Level of the Glutamine Synthetase Gene

As shown in Figure 13, the expression level of the *GS1* gene in the leaves of wt was lower at a nitrogen concentration of 0.1 mmol/L, and the difference between the overexpression and suppression lines was not significant. Under a nitrogen concentration of 2 mmol/L, the expression of o2 and i10 was lower, and the expression of i5 was the highest. The expression of i5 was the lowest at the 10 mmol/L nitrogen concentration, and the expression levels of o4 and i10 were higher. The expression of the *GS1* gene in the roots of wt and i5 was higher under the nitrogen concentration of 0.1 mmol/L, with the lowest expression in i10, and the difference between o2, o4, and i10 was not significant. The change trend under the condition of 2 mmol/L was essentially consistent with that under 0.1 mmol/L. At the nitrogen concentration of 10 mmol/L, the expression of o4 was the highest and that of i10 was the lowest, making them significantly different from the other strains.

The overexpression lines o2 and o4 showed high expression levels of the *GS2* gene in the leaves under the three nitrogen concentrations, i.e., 3–4 times higher than those of the other lines at 10 mmol/L. In contrast, wt exhibited lower expression levels at all three nitrogen concentrations. The o2 and o4 lines showed a high expression level of the *GS2* gene in the root at nitrogen concentrations of 0.1 mmol/L and 2 mmol/L. The expression levels of o2 and o4 were lower than those of wt, i5 and i10 under the condition of 10 mmol/L. The expression level of each strain increased as the nitrogen concentration increased.

Figure 14 shows the expression of the *Fd-GOGAT* gene in the leaves. The difference in each strain was not significant under a nitrogen concentration of 0.1 mmol/L. The expression of o2 and o4 was higher under nitrogen concentrations of 2 mmol/L and 10 mmol/L and was significantly different from wt, i5, and i10. There was no difference between wt, i5, and i10 at a 10 mmol/L nitrogen concentration.

At the three nitrogen concentrations, both o2 and o4 showed higher expression levels of the *Fd-GOGAT* gene in the roots, especially at the 10 mmol/L nitrogen concentration. It is worth noting that the i5 strain exhibited a high expression level at a nitrogen concentration of 2 mmol/L.

The expression level of the *NADH-GOGAT* gene in the leaves of the overexpression lines was higher at a nitrogen concentration of 0.1 mmol/L. Under a nitrogen concentration of 2 mmol/L, the expression level of i5 was higher, and the difference between the other strains was not significant. The expression of each strain was significantly increased under a nitrogen concentration of 10 mmol/L, and the expression of wt was lower.

The expression of the *NADH-GOGAT* gene in the roots of o2 was the highest at a nitrogen concentration of 0.1 mmol/L; there was no difference between wt and i5, which had low expression levels. The expression of i5 was the highest at the 2 mmol/L nitrogen concentration, and there was no difference between the other strains. The expression of wt was significantly increased at a nitrogen concentration of 10 mmol/L, and there was no difference between the other strains.

#### 2.5.10. Relative Expression Levels of the Nitrate Reductase and Nitrite Reductase Genes

As shown in Figure 15, the expression of the *NR* gene in the leaves of o2 was the highest under the nitrogen concentration of 0.1 mmol/L, the expression of o4 was the highest under the nitrogen concentration of 2 mmol/L, and the difference between the other strains was not significant. The expression was highest under 10 mmol/L nitrogen, and the expression of i10 was the lowest.

The overexpression lines o2 and o4 showed a high–low–high trend of expression of the *NR* gene in the leaves with increasing nitrogen concentration. There was no difference between the wt and suppression strains at 0.1 mmol/L and 2 mmol/L nitrogen concentrations. The expression of o4 was the highest under 0.1 mmol/L and 10 mmol/L. At 2 mmol/L nitrogen, o2 and o4 showed lower expression than the other strains.

The expression of the *NiR* gene in the leaves of o4 was higher at a nitrogen concentration of 0.1 mmol/L, and there was no difference between the other lines. The expression levels of o2 and o4 were higher than those of other strains at 2 mmol/L and 10 mmol/L nitrogen concentrations, and the difference was significant.

There was no difference in the expression of the *NiR* gene in the roots of each strain under the nitrogen concentration of 0.1 mmol/L. The expression of o2 and o4 was higher than that of the other strains at 2 mmol/L and 10 mmol/L nitrogen concentrations, and there was no difference between wt, i5, and i10.

## 3. Discussion

*PtrAMT1;6*, a high-affinity ammonium transporter protein member of *P. trichocarpa*, is homologous to tomato *LeAMT1;3*, and based on reported results for tomato *LeAMT1;3* [19], we hypothesized that *PtrAMT1;6* is associated with ammonium recycling in the aboveground parts. Thus, whether overexpression of the ammonium transporter protein *PtrAMT1;6*, which has an ammonium recycling function in the aboveground parts and no function in the roots, in poplar can change the biomass or wood quality of poplar was the question we wanted to investigate.

The *P. trichocarpa* genome contains 14 putative *AMT* genes, which is more than twice the number of *AMTs* in *Arabidopsis*. In the roots, *AMT1;2* is strongly expressed, *AMT1;6* is strongly induced by the circadian cycle, *AMT3;1* is expressed only in senescing poplar leaves, *AMT2;1* is highly expressed in leaves, and *AMT2;2* is mainly expressed in petioles. The specific expression of *AMT1;5* and *AMT1;6* in stamens and female flowers, respectively, suggests that they have key functions in the development of reproductive organs in poplar [5]. The results of the gene expression pattern analysis showed that *PtrAMT1;6* was mainly expressed in the leaves, and the transcript levels were higher in the mature leaves, petioles, and stems than in the young parts because catabolism in mature leaves is higher than that of young leaves, so we speculated that PtrAMT1;6 might be involved in the uptake of ammonium produced by metabolism in leaves and the reabsorption of ammonium produced during amino acid metabolism.

From sequence analysis, we can speculate that *PtrAMT1;6* is a membrane protein with 11 putative transmembrane structural domains and high homology with the N-terminal sequence of the tomato *LeAMT1;3* protein, both of which are relatively short. This finding predicts structural and functional homology between the two genes. Additionally, whether *PtrAMT1;6* has an ammonium transporter function due to its missing N-terminal sequence is a prerequisite for further studies.

The ammonium transporter protein gene was identified in yeast [20] and *Arabidopsis* [21] by functional complementation of yeast mutants. *AtAMT1;2* expressed in the yeast mutant exhibited biphasic kinetic properties (Km values of 36 μM and 3.0 mM) for methylammonium uptake [22]. *PtrAMT1;6* in this assay functioned to restore ammonium transport in the mutant, which is similar to the findings of others [23,24], suggesting that *PtrAMT1;6* has the ability to take up transported ammonium.

After the transformation of *Agrobacterium tumefaciens* GV3101-pCAMBIA 1301-*PtrAMT1;6P* into the *Arabidopsis* Col-0 ecotype, pure lines were screened and used for GUS staining. The results showed that, at the seedling stage, GUS was expressed mainly on the rootstock junction, cotyledon petioles, and the veins and mesophyll cells closely connected to the petioles, but not in the roots or the newly emerged leaves, whereas the AMT gene was expressed mainly in the roots of *Carya illinoinensis*, probably because *PtrAMT1;6* is a gene that controls the ammonium recycling function in the aboveground parts [25]. Although *Arabidopsis* and poplar are far apart in morphological structure, the expression activity of the *PtrAMT1;6* promoter in *Arabidopsis* is not fully indicative of its expression activity in *P. trichocarpa*, but this result suggests that the *PtrAMT1;6* promoter has promoter activity. To more accurately reveal the function of the *PtrAMT1;6* gene in *P. trichocarpa*, we selected mature ‘84K’ poplar for genetic transformation to obtain overexpression and suppression strains and performed an assay of relevant growth physiological indicators.

It has been reported that overexpressed *AMT* in rice led to an increased uptake rate at different nitrogen levels, but early overall biomass decreased, which the authors explained by indicating that the increased uptake rate was not followed by a corresponding increase in assimilation capacity [26], leading to ammonium toxicity [27]. In the present experiment, the two overexpression lines, o2 and o4, showed poor growth phenotypes, and the height, ground diameter, root length, and dry weight of the overexpression lines were lower than those of the suppressor and wt plants under the three nitrogen concentration treatments (0.1 mM, 2 mM, and 10 mM). Surprisingly, both suppressor lines showed good growth.

C and N metabolism in cells are inextricably linked, and their mutual coordination is essential for plant growth. Imbalances in carbon and nitrogen metabolism can lead to poor plant growth and decreased biomass [28]. The analysis revealed that overexpression of the *PtrAMT1;6* gene had little effect on C but increased the nitrogen content in the leaves, especially the leaves under low-nitrogen conditions, with the greatest increase, while AMT1 family genes are upregulated in plants under a low nitrogen environment [29], together with the effect of exogenous genes, thus altering the carbon and nitrogen metabolic state of ‘84K’ poplar and leading to imbalance in C and N metabolism, which is a possible plant growth poor cause. Bao A and Liang Z showed that overexpression of *OSAMT1;3* caused a significant reduction in both the C and N content of rice leaves, a higher leaf C/N ratio, and lower biomass in transgenic rice than in wt plants, which the authors analyzed as a result of the imbalance in C and N metabolism [30].

The content of soluble protein and soluble sugars reflect the strength of the nitrogen assimilation ability [31], and the overexpression plants had lower levels of soluble protein and soluble sugars than the suppressor and wt plants overall, especially in the stems and roots, which also indicated the lower nitrogen assimilation ability of the overexpression strain, and the glutamine synthetase content and free amino acid content also indicated the same. It is noteworthy that although the overexpression strains grew poorly, they showed higher levels of total nitrogen, soluble protein and soluble sugars, GS enzyme activity, and free amino acids in the leaves when treated with low nitrogen, and this is consistent with the previous results that *PtrAMT1;6* is a high-affinity ammonium transporter protein gene that functions in plant leaves under a low-ammonium environment.

Gene expression can, to some extent, represent the genetic potential of plant growth, and its gene expression changes with changes in the nitrogen supply level [32]. With increasing nitrogen concentration, carbon and nitrogen metabolism genes were upregulated overall in the overexpression lines, and the gene expression of *PtrAMT1;6* was more than four times higher in the overexpression strain than in the wt and suppressor strains at the three nitrogen levels. Similarly, *GS2*, *Fd-GOGAT*, and *NIR* showed higher expression, which may be due to the upregulation of *PtrAMT1;6* gene expression caused by the coordinated effect between genes involved in carbon and nitrogen metabolism [30]. Overall, the expression of genes related to carbon and nitrogen metabolism was somewhat higher in the transgenic plants than in wt plants, but the transgenic plants had significantly lower biomass than the wt plants, which could be due to a feedback regulatory effect in the transgenic overexpression plants. Although the overexpression of *AMT1;6* was mediated by a strong constitutive promoter (35S) in the transgenic plants, the possibility that the plants were subjected to feedback regulation cannot be excluded because this gene vector contains the complete 5’ and 3’ UTR regions, and the possibility that any transcriptional or posttranscriptional regulation occurred is high. Additionally, it is possible that posttranslational regulation leads to a decrease in transporter activity [27]. It has also been shown that N-terminal cysteine metastable regulation can affect the stability of the ammonium transporter protein *LeAMT1-1* oligomer [19]. Therefore, the posttranscriptional and posttranslational regulatory mechanisms of plant *AMT* cannot be neglected in future studies.

## 4. Materials and Methods

### 4.1. Plant Materials

*Populus trichocarpa*, *Arabidopsis thaliana*, and *Populus alba × P. glandulosa cv.* ‘84K’ were obtained from the germplasm resources of the National Key Laboratory of Forest Genetic Breeding, Northeastern Forestry University.

### 4.2. Strains, Vectors, and Reagents

The pEASY flat-end cloning vector and Trans1-T1 *E. coli* receptor cells were purchased from Beijing Alltech Biotechnology Co. (Beijing, China). The yeast expression vector pYES2 was purchased from Invitrogen Biotechnology. The yeast ammonium transporter mutant strain was 31019b (Δmep1, Δmep2, and Δmep3). The pROKEII plant expression vector and *Agrobacterium* strain GV3101 were obtained from the National Key Laboratory of Forest Genetic Breeding, Northeast Forestry University. pHANNIBAL was used as the RNAi intermediate vector, and pCAMBIA 2 was used as the RNAi plant expression vector.

The restriction endonucleases *Xho*I, *Hind*III, *Xba*I, and *Sac*I were purchased from TAKARA, and *BamH*I, *Kpn*I, *Sma*I, and *Bgl*II were purchased from NEB. The DNA polymerase used for common PCR identification was EasyTaq DNA polymerase produced by All Style Gold Biotechnology Co. (Beijing, China). The high-fidelity DNA polymerase used was KOD Plus Neo, produced by Toyobo Biotechnology Co. (Beijing, China).

The RNA extraction solution used was the pBIOZOL plant total RNA extraction kit produced by Beijing Baiyi Xinchuang Technology Co. (Beijing, China). The kits for quantitative PCR and plasmid extraction were produced by Beijing Kangwei Century Biotechnology Co. (Beijing, China). Free amino acids, glutamine synthetase, soluble sugars, and soluble proteins were determined using kits from Kemin Suzhou. Other reagents were made in China with analytical purity.

### 4.3. Sample Treatment for the Expression Pattern Analysis of the P. trichocarpa Ammonium Transporter Protein PtrAMT1;6

One-month-old histoculture seedlings were removed, and samples for photoperiodic expression pattern analysis were collected one month after seedling refinement, starting from the first fully expanded functional leaf and counting down three leaves. Each sample was a mixture of three *P. trichocarpa* plants. The collection period was divided into nine time points, 6:00, 7:00, 10:00, 14:00, 18:00, 22:00, 23:00, 2:00, and 5:00, and the greenhouse lights were switched on at 6:00 and off at 22:00. After two months of refinement, the samples were processed for whole-tissue expression pattern analysis: starting from the upper end of the morphology of *P. trichocarpa*, every three leaves were collected as one unit (the terminal bud was calculated as the first leaf but not collected), and groups of three leaves, petioles, and stems were collected. Six units were collected from each plant, and three mixed samples of *P. trichocarpa* were collected from each part. The main and lateral roots were collected separately.

### 4.4. Expression Pattern Analysis of Ammonium P. trichocarpa Transporter Protein PtrAMT1;6 Sample Treatment

To explore the similarity in the primary structure of the proteins of the members of *P. trichocarpa* and *Solanum lycopersicum* AMT1, we counted the protein sequences of the AMT1 family members of these two species and subjected the nitrogen-terminal sequences to multiple comparisons. The Phytozome site accession numbers of the members of hairy poplar *PtrAMT1;1-1;7* were Potri.010G063500, Potri.019G023600, Potri.008G173800, Potri.002G255100, Potri.002G255000, Potri.009G045200, and Potri.013G049600, and the sequence of *LeAMT1;1-1;3* was obtained from the literature [13]. Protein sequences were counted and then multiplexed using BioEdit 7.0.9.0 software. Transmembrane helices prediction of the PtrAMT1;6 protein was performed using the online software TMHMM: https://services.healthtech.dtu.dk/services/TMHMM-2.0/, accessed on 1 January 2016.

### 4.5. Cloning of the PtrAMT1;6 Gene and Construction of pEASY- PtrAMT1;6 in P. trichocarpa

The coding sequence of the PtrAMT1;6 gene (Porti.009G045200), with a total length of 1398 bp, was downloaded from Phytozome. The primers pEASY-S/A were designed, and the target fragment was amplified by RT–PCR, using cDNA of P. trichocarpa leaves as a template. The product was ligated into the pEASY-Blunt Cloning Vector. See Appendix A for the primers.

### 4.6. pYES2-AMT1;6 Vector Construction and Functional Complementation in Yeast

Primers pYES2-S/A containing *Xba*I and *Sac*I restriction sites were designed. pEASY-*PtrAMT1;6* plasmid DNA was used as a template for PCR amplification and double digestion to construct the recombinant plasmid pYES2-Ptr AMT1;6. Yeast was streaked onto SC-U plates, and a single clone was picked and grown in SC-U liquid medium and incubated at 30 °C for 2 days to an OD_λ = 600_ of 0.5. We grew 5 μL spots of each of the five concentrations (10^−1^, 10^−2^, 10^−3^, 10^−4^, and 10^−5^) on media containing 0.4 mM, 1 mM, 20 mM, and 100 mM ammonium and 0, 10, 50, and 100 mM MEA for 2–4 days.

### 4.7. PtrAMT1;6P::GUS Construction and Expression in Arabidopsis

#### 4.7.1. Promoter Element Analysis

The promoter sequences were entered into the PlantCARE website, which generates a list of cis-acting elements to annotate the relevant elements (http://bioinformatics.psb.ugent.be/webtools/plantcare/html/, accessed on 1 January 2016.).

#### 4.7.2. Cloning and Expression Vector Construction of the PtrAMT1;6 Promoter in Populus Tridentata

The coding sequence of the *PtrAMT1;6* (Potri.009G045200) gene was downloaded from the Phytozome website upstream of 2134 bp as the promoter sequence; the primer AMT1;6-S/A was designed; the target fragment was amplified by PCR, using the genomic DNA of *P. trichocarpa* as the template; and the product was ligated into Trans1-T1. The constructed pCAMBIA1301-PtrAMT1;6 plasmid was transformed into *Agrobacterium tumefaciens* GV3101, and the strain was stored at −80 °C. See Appendix A for the primers.

#### 4.7.3. Construction of the Genetic Transformation System and GUS Staining in Arabidopsis

The grown *Arabidopsis thaliana* Col-0 ecotype was genetically transformed by the *Agrobacterium*-mediated flower-dipping method, and the seeds of the obtained T2-generation pure lines were cultured aseptically for 7 days and then stained for GUS.

### 4.8. Determination of Genetic Transformation and Physiological Data of the ‘84K’ Poplar PtrAMT1;6 Gene

#### 4.8.1. Carrier Construction

The PART1 fragment was amplified with the recombinant plasmid pEASY-PART1 as a template, and the plasmid of the intermediate vector pHANNIBAL was extracted. The plasmid and PCR product were treated with restriction endonuclease, the PART2 fragment was amplified with the recombinant plasmid pEASY-PART1 as a template, and the plasmid of the recombinant vector pHANNIBAL-PART1 was extracted. The purified DNA fragment was sticky-end ligated, and the plasmid was extracted by transforming *E. coli*. The recombinant plasmid pCAMBIA 2300-RNAi was constructed by transforming *Agrobacterium tumefaciens* EHA105 with pHANNIBAL-RNAi and pCAMBIA 2300 plasmids, and the strain was stored at −80 °C for safekeeping.

#### 4.8.2. Construction of ‘84K’ poplar PtrAMT1;6 Gene Overexpression and Repressed Expression Strains

Two genetic transformation systems were obtained by *Agrobacterium*-mediated genetic transformation of ‘84K’ poplar wt histoculture seedling leaves with the preserved EHA105-pROKII- *PtrAMT1;6* and EHA105-pCAMBIA 2300-RNAi strains. See Appendix A for the primers.

#### 4.8.3. Physiological Index Measurement

The ‘84k’ poplar group seedlings were cultured in rooting medium for one month, and then healthy and uniformly growing seedlings were selected for refining. Hoagland nutrient solution (2 mM NH_4_Cl) was cultured for one week and then weaned from nitrogen for 3 days, after which the nutrient solution was replaced every three days, and samples were collected beginning after 2 weeks of culture. Starting from the first fully expanded leaf at the upper end of the ‘84k’ poplar morphology, every three leaves were collected as a unit, and the leaves, stems, and roots were snap-frozen in liquid nitrogen and stored at −80 °C for various growth and physiological index determinations. The remaining plants were cultured in the greenhouse for one month, and after two weeks of treatment with different nitrogen concentrations, their photosynthetic parameters and chlorophyll content were measured using the LI-6400 photosynthesis system and SPAD-502 chlorophyll meter.

## 5. Conclusions

We verified by functional complementation of yeast mutants and GUS staining of promoter mutants that *PtrAMT1;6* is an aboveground partly acting gene with ammonium uptake transporter ability and that the promoter has promoter activity. We also examined the physiological indicators and gene expression levels of transgenic ‘84k’ poplar under different ammonium treatments and concluded that *PtrAMT1;6* is a high-affinity ammonium transporter protein gene that functions in plant leaves under a low-ammonium environment and that upregulation of *PtrAMT1;6* gene expression may cause a coordinated interaction between genes involved in carbon and nitrogen metabolism. This study provides a basis for the further identification of the functions of *AMT* genes in *P. trichocarpa* and provides a theoretical basis for biomass or wood quality improvement in *P. trichocarpa*.

## Figures and Tables

**Figure 1 ijms-24-08511-f001:**
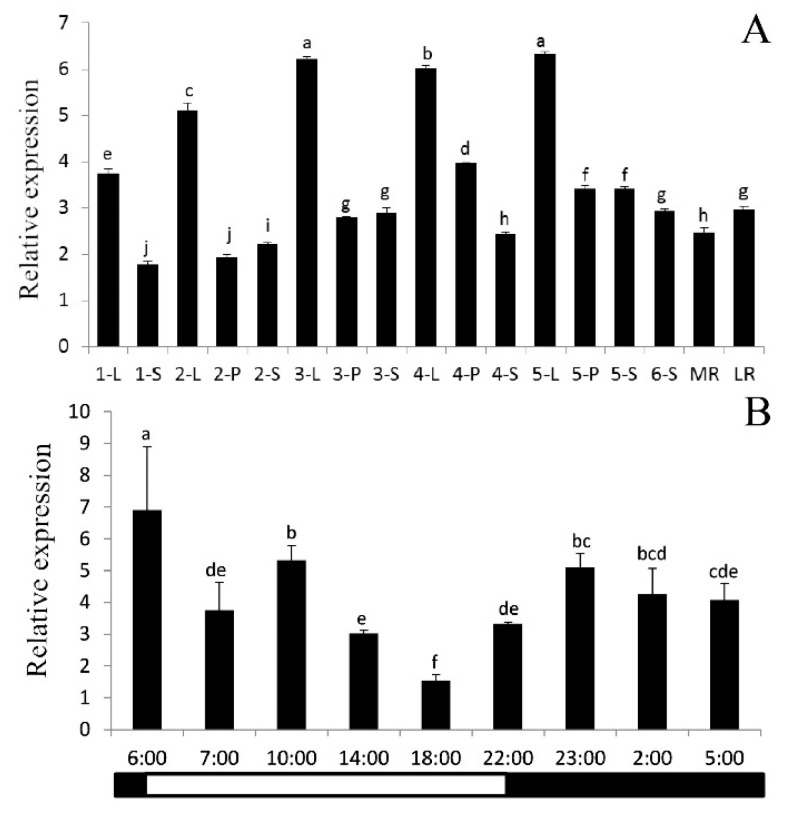
PtrAMT1;6 gene expression measured by real-time quantitative PCR. (**A**) Relative expression of the ammonium transporter protein *PtrAMT1;6* gene in various tissue sites in *P. trichocarpa* (L indicates leaf; S indicates stem; P indicates petiole; MR indicates main root; LR indicates lateral root; numbers represent the youngness of the tissue (see Appendix A for details)). (**B**) Photoperiodic expression pattern of the *PtrAMT1;6* gene in *P. trichocarpa* leaves. The *p* < 0.05 significance levels are indicated by different letters, and the data are the means ± SEs of three biological and three technical replicates.

**Figure 2 ijms-24-08511-f002:**
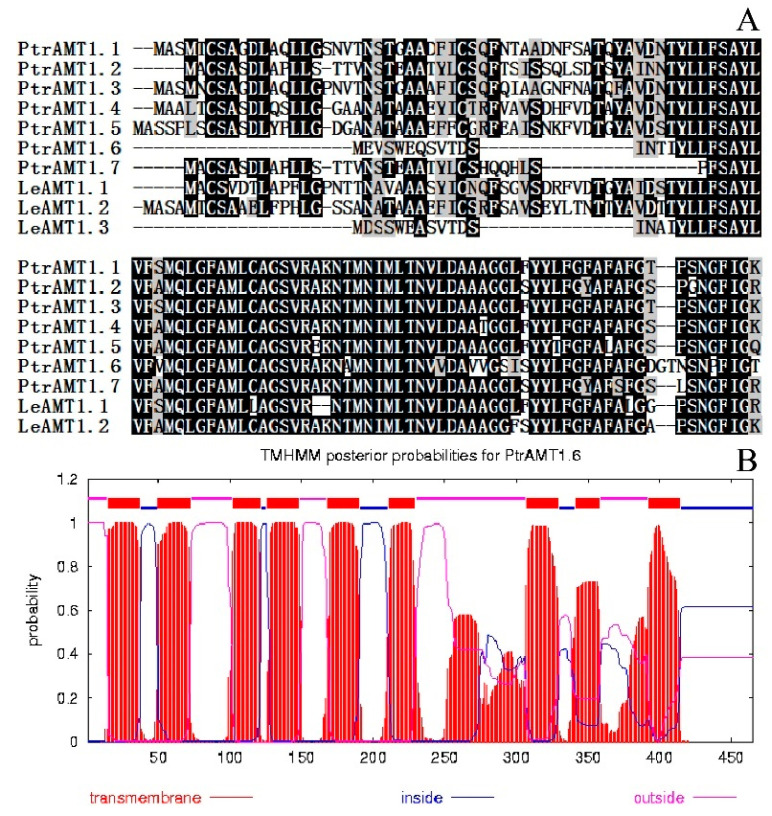
(**A**) Multiplexed sequence comparison of the protein N-terminal sequences of AMT family members from *P. trichocarpa* and *S. lycopersicum*. Black indicates identical sequences; white indicates different sequences. (**B**) Predicted transmembrane helices of the ammonium transport protein PtrAMT1;6.

**Figure 3 ijms-24-08511-f003:**
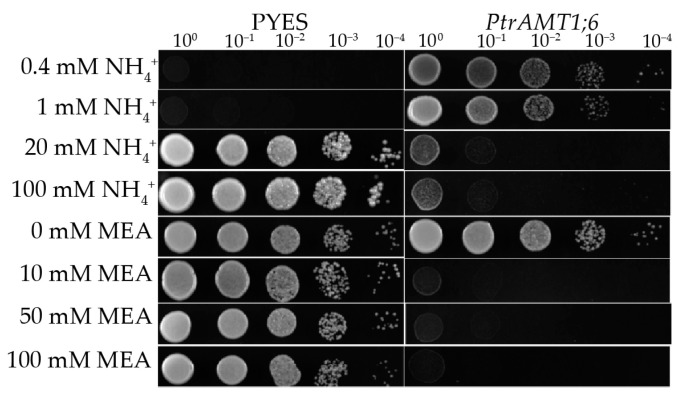
Functional complementation assay of a yeast ammonium transporter protein mutant. Growth experiments were performed on media containing different concentrations of NH^+^ and MEA as the sole source of nitrogen. Photographs were taken after two days of incubation at 30 °C.

**Figure 4 ijms-24-08511-f004:**
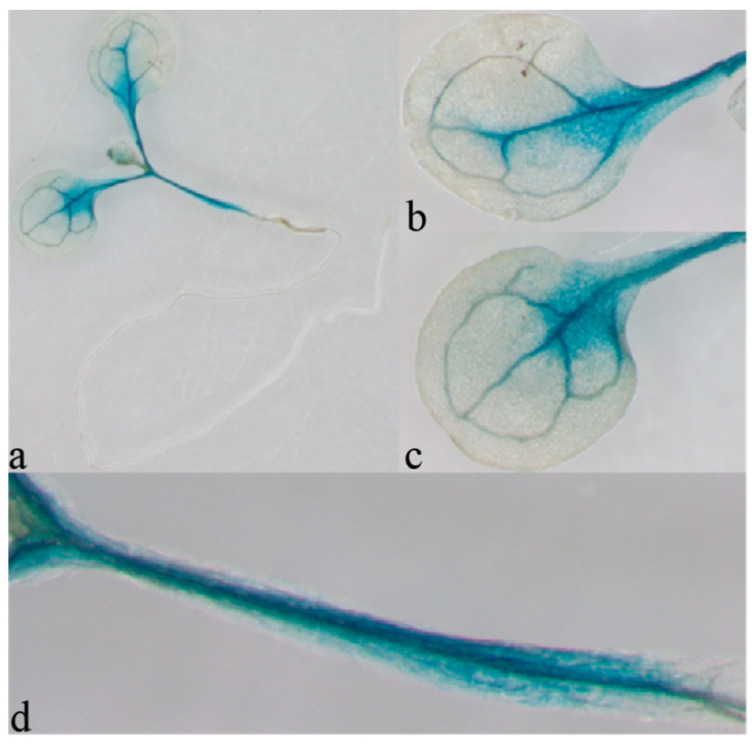
*PtrAMT1;6*P::GUS expression in different parts of Arabidopsis: (**a**) whole seedlings, (**b**) right cotyledon, (**c**) left cotyledon, and (**d**) rootstock junction.

**Figure 5 ijms-24-08511-f005:**
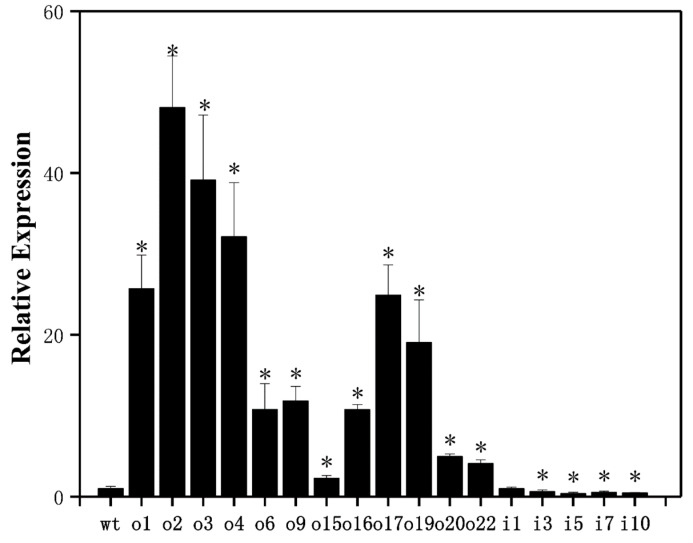
Overexpression/suppression lines and wt plants were tested for relative expression of the *ptrAMT1;6* gene by RT–PCR under normal growth conditions, and the significance level of *p* < 0.05 is indicated by *. The data are the means ± SEs of three biological and three technical replicates.

**Figure 6 ijms-24-08511-f006:**
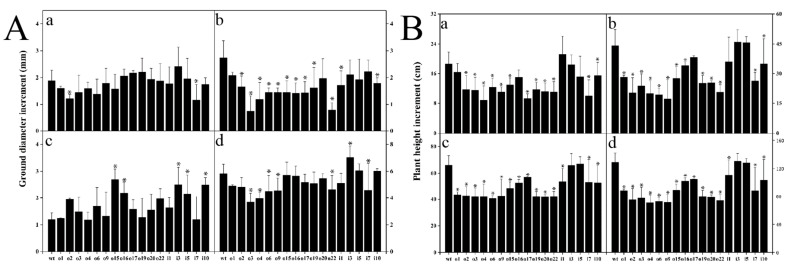
(**A**) Plant height increase in overexpression/suppression lines versus wt plants after 60 days of growth in soil culture in a greenhouse. (**B**) Increase in the ground diameter of overexpression/suppression lines and wt plants after 60 days of growth in soil culture in greenhouses. Letters a, b, c, and d represent increases from 0 to 20 d, 20 to 40 d, 40 to 60 d, and 0 to 60 d, respectively; significance levels of *p* < 0.05 are indicated by *, and the data are the means ± SEs of three biological and three technical replicates.

**Figure 7 ijms-24-08511-f007:**
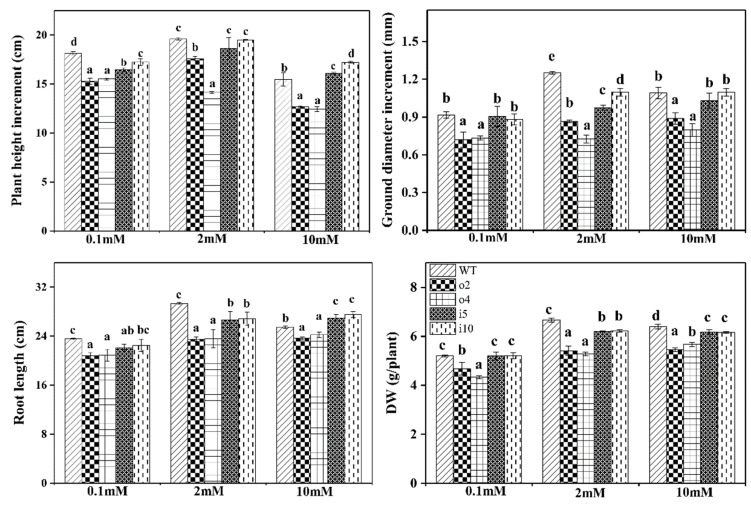
The overexpression/suppression lines and wt plants were cultured for 14 days at three NH_4_Cl concentrations, and plant height, ground diameter, root length, and dry weight (DW) were measured. The significance level of *p* < 0.05 is indicated by different letters, and the data are the means ± SEs of three biological and three technical replicates.

**Figure 8 ijms-24-08511-f008:**
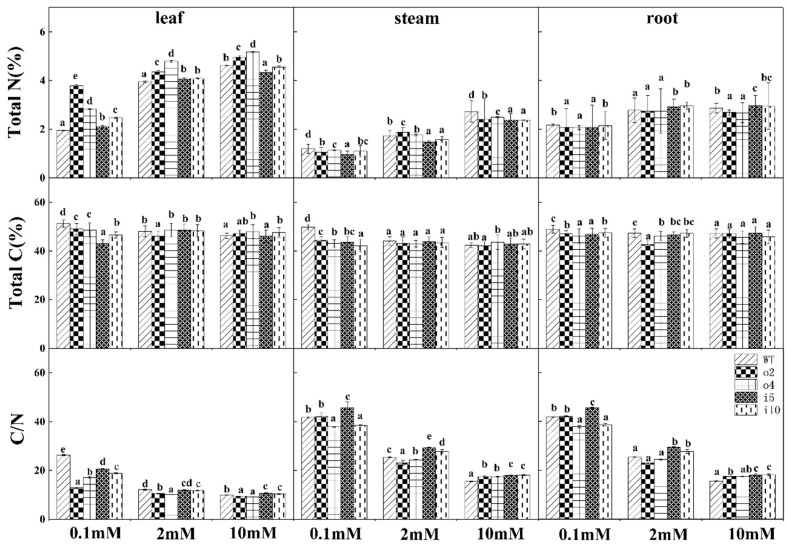
The carbon and nitrogen contents in the leaves, stems, and roots of overexpression/suppression strains and wt plants were examined after 14 days of incubation at three NH_4_Cl concentrations, and their carbon-to-nitrogen ratios were calculated. The *p* < 0.05 significance levels are indicated by different letters, and the data are the means ± SEs of three biological and three technical replicates.

**Figure 9 ijms-24-08511-f009:**
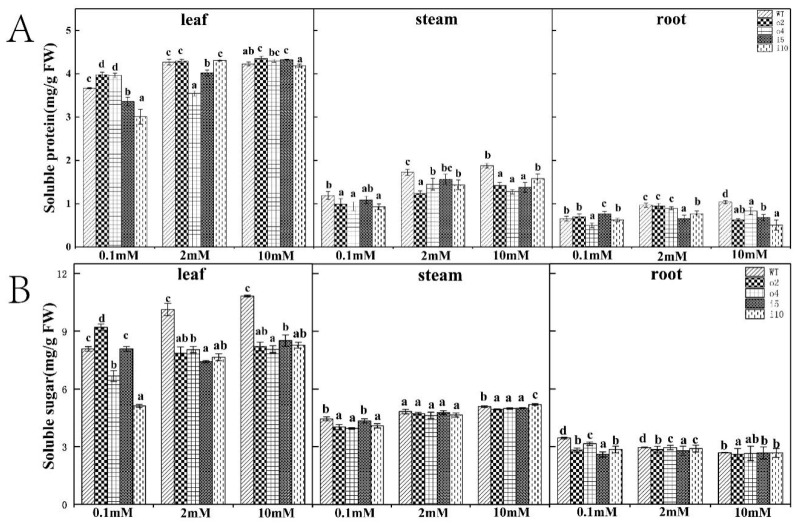
Soluble protein (**A**) and soluble sugar contents (**B**) in the leaves, stems, and roots of the overexpression/suppression strains and wt plants were examined after 14 days of incubation at three NH_4_Cl concentrations. The *p* < 0.05 significance levels are indicated by different letters, and the data are the means ± SEs of three biological and three technical replicates.

**Figure 10 ijms-24-08511-f010:**
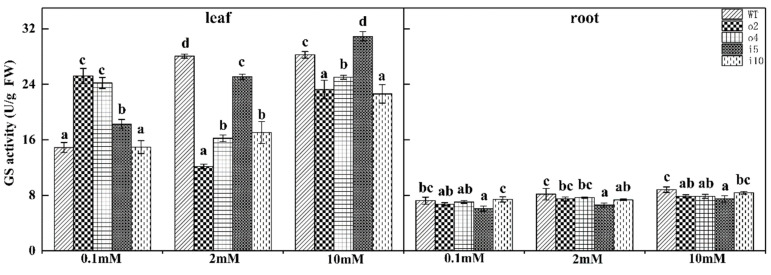
Overexpression/suppression lines and wt plants were tested for GS activity in leaves and roots after 14 days of culture at three NH_4_Cl concentrations. The significance level of *p* < 0.05 is indicated by different letters, and the data are the means ± SEs of three biological and three technical replicates.

**Figure 11 ijms-24-08511-f011:**
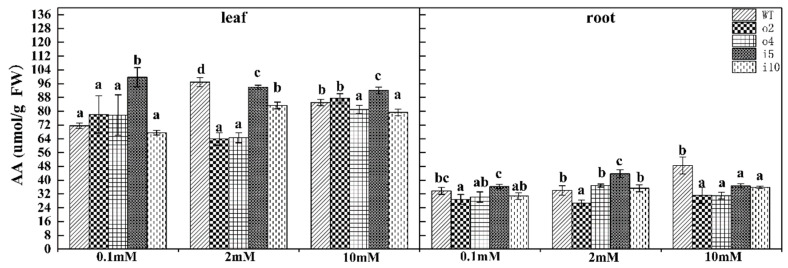
Overexpression/suppression lines and wt plants were tested for free amino acid content in the leaves and roots after 14 d of culture at three NH_4_Cl concentrations. The significance level of *p* < 0.05 is indicated by different letters, and the data are the means ± SEs of three biological and three technical replicates.

**Figure 12 ijms-24-08511-f012:**
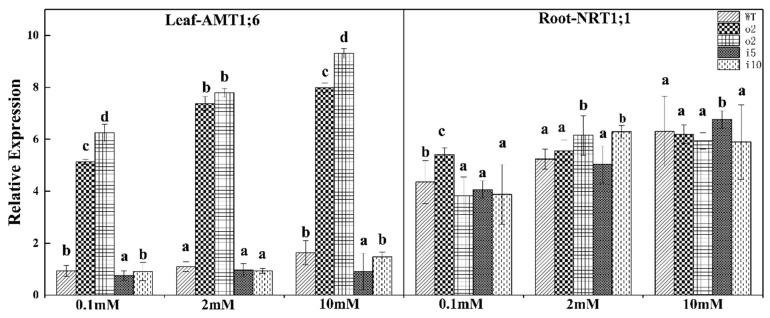
Overexpression/suppression lines were incubated with wt plants at three NH_4_Cl concentrations (0.1 mM, 2 mM, and 10 mM) for 14 days and then subjected to RT–PCR for *AMT1;6* in leaves and *NRT1;1* in roots. The *p* < 0.05 significance levels are indicated by different letters, and the data are the means ± SEs of three biological and three technical replicates.

**Figure 13 ijms-24-08511-f013:**
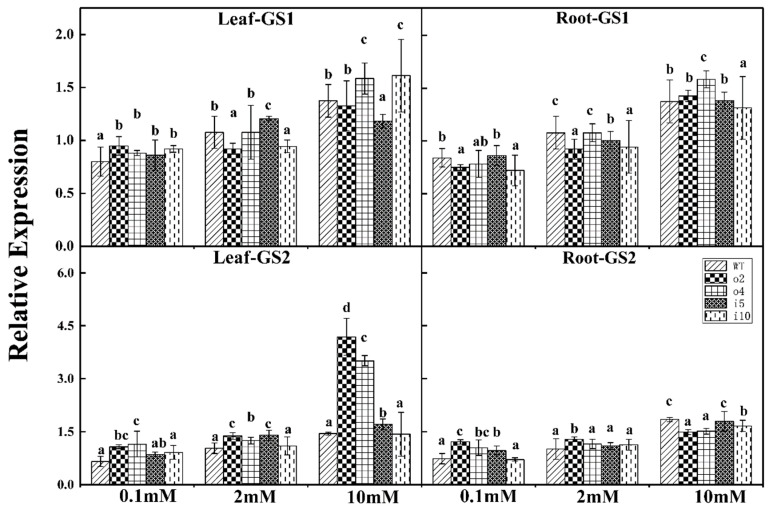
Overexpression/suppression plants and wt plants were subjected to RT–PCR for *GS1* and *GS2* in roots and leaves after 14 days of incubation at three NH_4_Cl concentrations (0.1 mM, 2 mM, and 10 mM). The *p* < 0.05 significance levels are indicated by different letters, and the data are the means ± SEs of three biological and three technical replicates.

**Figure 14 ijms-24-08511-f014:**
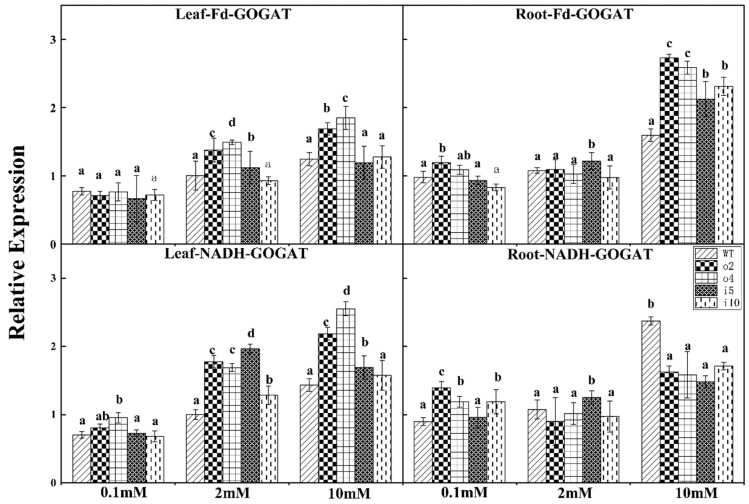
RT–PCR was performed for *Fd-GOGAT* and *NADH-GOGAT* in roots and leaves after 14 days of incubation at three NH_4_Cl concentrations (0.1 mM, 2 mM, and 10 mM) in overexpression/suppression strains and wt plants. The *p* < 0.05 significance levels are indicated by different letters, and the data are the means ± SEs of three biological and three technical replicates.

**Figure 15 ijms-24-08511-f015:**
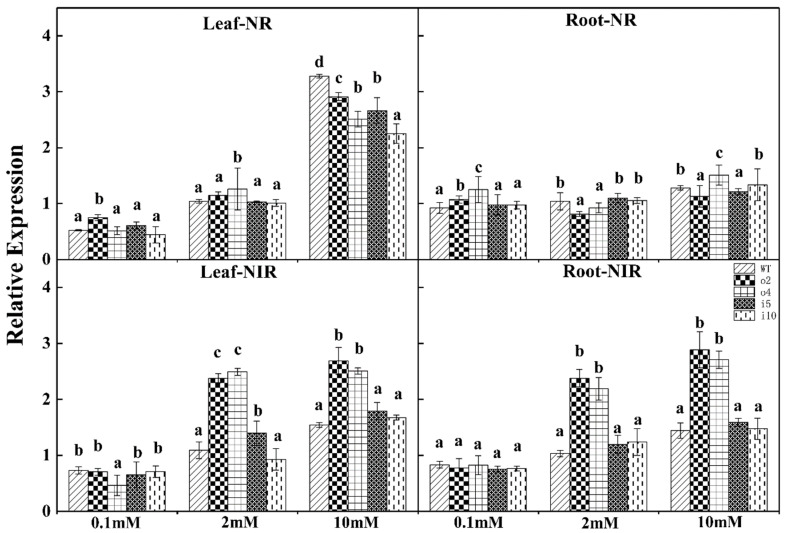
Overexpression/suppression plants and wt plants were subjected to RT–PCR for *NR* and *NIR* in roots and leaves after 14 days of incubation at three NH_4_Cl concentrations (0.1 mM, 2 mM, and 10 mM). The *p* < 0.05 significance levels are indicated by different letters, and the data are the means ± SEs of three biological and three technical replicates.

**Table 1 ijms-24-08511-t001:** Promoter subcomponents.

Name	Position	Strand	Function
G-box	417	−	Light-responsive element
ATCT-motif	1069	+	Light-responsive element
Box 4	592	+	Light-responsive element
CATT-motif	691	−	Light-responsive element
CG-motif	602	+	Light-responsive element
GA-motif	247	−	Light-responsive element
Box I	9	−	Light-responsive element
Box I	540	−	Light-responsive element
Sp1	859	−	Light-responsive element
ERE	540	−	Ethylene-responsive element
circadian	804	+	Circadian control
circadian	976	+	Circadian control
ARE	902	−	Anaerobic induction
ARE	1192	+	Anaerobic induction

**Table 2 ijms-24-08511-t002:** Photosynthetic parameters of the overexpression/suppression lines and wt plants after 30 days in greenhouse soil culture.

Strain	SPAD	Photosynthetic Rate (μmol CO_2_·m^−2^·s^−2^)	Stomatal Conductance (mmol·m^−2^·s^−1^)	Intercellular Carbon Dioxide Concentration (μL·L^−1^)	Transpiration Rate (mmol H_2_O·m^−2^·s^−1^)
wt	12.5 ± 0.1	11.28 ± 1.83	0.66 ± 0.07	179.8 ± 38.55	4.34 ± 0.45
O1	8.97 ± 0.12 *^1^	9.36 ± 0.76	0.66 ± 0.03	156.7 ± 27.95 *	5.26 ± 0.14
O2	8.47 ± 0.06 *	8.68 ± 1.82 *	0.71 ± 0.08	159.7 ± 40.77 *	5.56 ± 0.55 *
O3	8.1 ± 0.06 *	8.63 ± 1.59 *	0.66 ± 0.06	173.4 ± 13.55	5.15 ± 0.38
O4	9.57 ± 0.15 *	10.2 ± 0.97	0.75 ± 0.05	146.2 ± 13.77 *	5.9 ± 0.31 *
O6	8.6 ± 0.12 *	8.27 ± 2.13 *	0.59 ± 0.04	173.7 ± 36.26	4.92 ± 0.31
O9	9.57 ± 0.37 *	8.6 ± 1.31 *	0.71 ± 0.05	168.5 ± 23.29	5.66 ± 0.34 *
O15	9.77 ± 0.07 *	11.18 ± 1.75	0.73 ± 0.06	150.49 ± 9.39 *	5.7 ± 0.5 *
O16	7.37 ± 0.1 *	7.71 ± 1.11 *	0.68 ± 0.04	213.8 ± 26.56	2.91 ± 0.18 *
O17	7.73 ± 0.15 *	7.84 ± 0.59 *	0.65 ± 0.01	200.6 ± 19.05	3.62 ± 1.34
O19	8.8 ± 0.25 *	10.03 ± 0.97	0.7 ± 0.03	136.6 ± 12.54 *	5.72 ± 0.19 *
O20	7.83 ± 0.2 *	7.61 ± 0.58 *	0.72 ± 0.04	223.5 ± 7.57	3.2 ± 0.66 *
O22	9.97 ± 0.25 *	10.78 ± 1.1	0.75 ± 0.05	165.5 ± 11.65	5.63 ± 0.82 *
i1	11.83 ± 0.15	11.98 ± 1.02	0.71 ± 0.15	193.0 ± 40.63	4.81 ± 0.91
i3	10.27 ± 0.17 *	11.47 ± 2.23	0.56 ± 0.13 *	176.3 ± 25.45	4.44 ± 1.1
i5	10.77 ± 0.21	11.84 ± 3.03	0.53 ± 0.19 *	200.2 ± 27.54	4.33 ± 1.61
i7	8.73 ± 0.15 *	9.96 ± 1.14	0.46 ± 0.05 *	200.62 ± 29.5	3.71 ± 0.35 *
i10	9.73 ± 0.15 *	11.96 ± 0.2	0.77 ± 0.01	186.0 ± 49.97	5.89 ± 0.27 *

^1^ Significance levels of *p* < 0.05 are indicated by *, and the data are the means ± SEs of three biological and three technical replicates.

**Table 3 ijms-24-08511-t003:** Detection of photosynthetic parameters after overexpression/suppression of wt plants in wt plants after 14 days at three NH_4_Cl concentrations.

Concentration	Strain	SPAD	Photosynthetic Rate (μmol CO_2_·m^−2^·s^−2^)	Stomatal Conductance (mmol·m^−2^·s^−1^)	Intercellular Carbon Dioxide Concentration (μL·L^−1^)	Transpiration Rate (mmol H_2_O·m^−2^·s^−1^)
0.1 mM	WT	8.37 ± 0.21 c^1^	12.68 ± 0.12 d	0.68 ± 0.01 d	218.75 ± 2.42 d	5.78 ± 0.11 c
	o2	7.67 ± 0.25 b	11.95 ± 0.02 bc	0.57 ± 0.03 b	197.17 ± 0.58 a	5.74 ± 0.05 c
	o4	7.63 ± 0.21 b	11.55 ± 0.04 a	0.67 ± 0.01 cd	201.89 ± 1.43 ab	5.55 ± 0.04 b
	i5	7.13 ± 0.15 a	12.1 ± 0.21 c	0.53 ± 0.01 a	205.46 ± 2.17 bc	5.37 ± 0.02 a
	i10	7.57 ± 0.15 b	11.83 ± 0.12 b	0.66 ± 0.02 c	209.35 ± 7.66 c	5.5 ± 0.01 b
2 mM	WT	10.57 ± 0.25 d	16.63 ± 0.23 d	0.75 ± 0.02 c	253.33 ± 0.94 b	6.17 ± 0.1 c
	o2	8.83 ± 0.06 bc	15.12 ± 0.2 b	0.71 ± 0.03 a	227.53 ± 8.57 a	5.92 ± 0.06 a
	o4	8.6 ± 0.1 ab	14.55 ± 0.04 a	0.72 ± 0.01 ab	233.09 ± 4.26 a	6.09 ± 0.02 b
	i5	8.37 ± 0.21 a	15.99 ± 0.04 c	0.76 ± 0.01 bc	224.31 ± 13.35 a	6.1 ± 0.01 b
	i10	9.1 ± 0.1 c	14.55 ± 0.04 a	0.7 ± 0.02 a	229.73 ± 1.7 a	6.05 ± 0.01 b
10 mM	WT	13.73 ± 0.15 d	19.95 ± 0.13 e	0.81 ± 0.07 d	284.07 ± 2.65 d	6.81 ± 0.01 d
	o2	9.5 ± 0.26 a	17.93 ± 0.1 b	0.76 ± 0.02 b	247.65 ± 1.42 a	6.14 ± 0.05 a
	o4	10.33 ± 0.15 b	17.31 ± 0.2 a	0.75 ± 0.01 b	258.71 ± 4.16 d	6.38 ± 0.06 c
	i5	11.47 ± 0.25 c	18.2 ± 0.09 c	0.73 ± 0.02 a	273.6 ± 1.22 c	6.28 ± 0.04 b
	i10	11.1 ± 0.3 c	18.65 ± 0.07 d	0.78 ± 0.04 c	284.17 ± 2.21 d	6.89 ± 0.2 e

^1^ Different letters indicate significant differences between the two (*p* < 0.05), and the data are the means ± SEs of three biological and three technical replicates.

## Data Availability

Not applicable.

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
