# Peer review of "Functional Identification and Genetic Transformation of the Ammonium Transporter PtrAMT1;6 in Populus"

_ijms, 2023, doi:10.3390/ijms24108511_

Round 1

Reviewer 1 Report

In this manuscript the authors characterized PtrAMT1;6 expression and function through various experimental means, and then examined the effects of its overexpression in poplar. My biggest issue for the authors is that several experiments in this manuscript were previously reported elsewhere, but the authors do not make that clear. In particular, the authors did not properly cite and acknowledge work from Couturier et al, 2007, which previously examined tissue-dependent expression of PtrAMT1;6 by RT-qPCR, as well as its complementation of yeast strain 31019b. Instead, Couturier et al was cited in line 64 only as having provided insights into tomato AMT function (LeAMT1;1, LeAMT1;2, and LeAMT1;3). However, Couturier et al in fact did many of the same experiments on poplar AMTs including AMT1;6:

-        Figure 2 in the Couturier paper reported the expression of PtrAMT1;6 primarily in the leaves by RT-qPCR. This compares to the Figure 1A in this manuscript.

-        Figure 5 of the Couturier paper reported the expression of PtrAMT1;6 while monitoring light/dark cycle. This compares to Figure 1B in this manuscript.

-        Figures 9 and 10 of the Couturier paper reported the functional complementation of yeast strain 31019b with ammonium- and methylammonium-supplemented media. These compare to Figure 3 in this manuscript.

The authors must properly cite prior work and acknowledge which experiments of theirs are novel and which are repeats of previously published work. This has severe implications for how the abstract, introduction, results and discussion sections are all written.

Beyond this critical issue, there are multiple issues with how data are presented in this manuscript that must be resolved:

Figure 1: It is not currently stated in either the accompanying results text or the figure caption what specific experiment was performed in Figure 1 to measure expression levels. It is presumably RT-qPCR, but it must be stated in the text and/or caption exactly what is being measured in this experiment. Likewise, when reporting relative expression levels the authors must define what they used as their baseline of 1. None of the bars are even with the value of 1, raising questions about what the authors used for their comparison. Also, the letters above each bar are never defined.

Figure 2: Numbering should be indicated to the right of each line for which amino acid position each line ends with. In the lower panel, the text says the server TMPRED was used but the figure says TMHMM was used. TMHMM is a different program, so this discrepancy must be resolved. Also, the authors should include the citation for the server used in the accompanying text on line 118.

Figure 3: The caption should state what fold dilutions the plating used and for how long the cells were grown before being imaged. How many times was this experiment performed? Are the images here representative? Also there are many instances throughout the manuscript where abbreviations are used without first identifying what they stand for. In this case, indicate what the alternative nitrogen source MEA stands for (line 149 and onward).

Figure 4: The caption must define each of panels (a), (b), (c), and (d).

Figure 5: In accompanying text line 206, give a rationale for why lines o2 and o4 were selected for further experimentation, as opposed to any of the other overexpression lines.

Smaller points:

Line 10: This should state that the “AMT family” are a group of important ammonium transporters, rather than treating the term AMT as though it were a single transporter.

Line 18: The phrase “the yeast ammonium transporter protein mutant 31019b mutant” is confusing. 31019b is a yeast strain with three ammonium transporters knocked out. Reword to make that clear.

Line 28: Rather than stating the overexpression “affected its normal growth and development,” be specific in stating what the effect is: that overexpression resulted in decreased biomass.

Line 28: The abstract would greatly benefit from a final sentence that synthesizes the results and states the main takeaway from the experiments described in the preceding 16 lines.

Lines 72-81 are an especially long single sentence and should be broken up for clarity.

Line 115 should say “multiple sequence alignment”, not “multiple alignment”

Line 117: What is the citation for the claim that cysteines stabilize trimers? A citation is needed here.

Line 119: The output of Tmpred or TMHMM is to predict TM helices, not TM domains. This wording should be changed throughout the text in all instances.

Line 149: define MEA. Furthermore, address all other instances of acronyms that are introduced without first defining what they are.

Line 168: refers to Fig. 6, but must mean Fig. 4 instead.

Reviewer 2 Report

The work is equipped with different methods aimed at understanding the use of ammonium.

My advice, in order to improve the quality of the paper, mainly concerns the correction of the abstract and add bibliographic reference, especially in the discussion.

In the introduction there are too many repetitions.

In the introductions some aspects are missing (see comments)

Reviewer 3 Report

This paper entitled “Functional identification and genetic transformation of the Ammonium transporter PtrAMT1;6 in Populus” showed that PtrAMT1;6 is an aboveground partly acting gene with ammonium uptake transporter ability and that the promoter has promoter activity. In addition, PtrAMT1;6 is a high-affinity ammo nium transporter protein gene that functions in plant leaves under a low-ammonium environment and that upregulation of PtrAMT1;6 gene expression may cause a coordinated interaction between genes involved in carbon and nitrogen metabolism. The topic was interesting and the paper was well-written. There are several minor problems to be addressed.

1, Fig.7 measured the DW of per plant, however, in fig.9, the concentrations of soluble protein and sugars were present with the FW, why?

2, In fig.3, what does “MEA”mean?

3, The format of references should be uniform, such the name of magazine, such as line 681 “Plant Signal. Behav”

line 689 “The New phytologist”

line 701 “Plant & cell physiology”

Capital or small letter? Full name or abbreviation?

Round 2

Reviewer 1 Report

The authors have addressed my comments.